# Preserving exposed hydrophilic bumps on multi-bioinspired slippery surface arrays unlocks high-efficiency fog collection and photocatalytic cleaning

Junda Wu, Chunxiang Li, Jiangdong Dai ✉ & Yan Yan ✉

The efficiency of fog collection technologies is inherently hindered by the long-standing dilemma of capture vs. transportation balance. Inspired by nature, we address this issue by preserving hydrophilic bumps on slippery liquid-infused porous surfaces (SLIPS) through an underwater infusion strategy, creating a super-slippery fog collector with multi-scale biomimetic structures. This surface combines features from beetle carapaces and pitcher plant surfaces, enabling rapid initial water capture on hydrophilic bumps and efficient droplet transport. As a result, we develop the most efficient fog-collecting surface reported to date, harvesting 5000-60000 mg/cm² per hour with fog flow rates ranging from 300-1500 mL/h. By macroscopically scaling and optimizing, we construct an integrated 3D fog collecting device capable of harvesting over 660 g of water in 500 minutes. Further integrating $TiO_2$ into the bumps imparts the ability for simultaneous water collection and purification without sacrificing collection efficiency. Our work reveals that resolving the capture-transport dichotomy is key to achieving high-efficiency fog water collection.

Water scarcity remains one of the most critical challenges facing the global population, particularly in arid and semi-arid regions where conventional water sources are often limited[1–3]. In these areas, atmospheric water in the form of fog represents an underexploited resource that could significantly alleviate water shortages if effectively collected[4]. The primary challenge of current fog harvesting systems lies in optimizing the capture vs. transportation balance of water droplets; these two processes are crucial for efficient water collection but are inherently contradictory in principle[5,6]. On conventional large mesh nets made of woven polymer[7,8], typically only one of these aspects can be optimized by adjusting the wettability, leaving it an unresolved dilemma.

Inspired by nature's unique structures and properties, innovative advancements in material surface and interface technologies have been made to enhance the water collection rate[9–14]. One such innovation mimics the carapace of desert beetles[15], where an interlaced configuration of hydrophilic microstructures can easily capture tiny fog droplets until they grow large enough to roll off. This surface Janus structure undoubtedly accelerates the initial capture of water droplets. However, its alternating hydrophilic and hydrophobic surface distribution creates adhesion that hinders droplet detachment and surface migration[16,17]. Consequently, desert beetles need to tilt its body towards to the wind flow and the water droplets with the diameter of critical size could be transported to their mouths. Interestingly, nature also provides solutions to the problem of droplet transportation, pointing towards the construction of slippery liquid-infused porous surfaces (SLIPS)[18], inspired by the water-slippery nature of peristome from the Nepenthes pitcher plant to hunt insects[19]. With only a certain

Institute of Green Chemistry and Chemical Technology, School of Chemistry and Chemical Engineering, Jiangsu University, Zhenjiang, China.
✉e-mail: daijd@ujs.edu.cn; dgy5212004@163.com

angle of inclination, little droplets can be easily slid, transported, and collected by SLIPS. However, constructing stable SLIPS necessitates a superhydrophobic substrate infused with inert hydrophobic lubricating oil[18], which creates a significant water coalescence barrier, thus constraining the whole water collection rate.

Given the limitations of these single bio-inspired structures in addressing either capture or transport, the key steps in fog harvesting, we propose that combining the characteristics of both structures could be an optimal solution. By constructing superhydrophobic SLIPS with exposed hydrophilic bumps on top, we can ideally balance the capture and transportation of water droplets (Fig. 1a). However, a significant challenge is that, according to surface energy theories[20], hydrophilic components are also inherently oleophilic in air. This means that during traditional lubricating oil infusion processes, the hydrophilic bumps are bound to be wrapped by oil. Thus, although much desired, achieving a stable superhydrophobic SLIPS with exposed hydrophilic top structures is extremely challenging.

In this work, we developed an underwater lubricating oil infusion strategy for multi-bio-inspired SLIPS, preserving exposed hydrophilic bumps on the hydrophobic lubricating substrate (as illustrated in Fig. 1b). This design facilitates rapid initial water capture at hydrophilic bumps, combined with the efficient sliding transport of the formed water droplets, effectively solving the longstanding dilemma between capture and droplet transport.

## Results

### Sample fabrication and characteristic analysis

Inspired by the hydrophilic/hydrophobic hybrid microstructure on the desert beetle back carapace, we designed a Janus structure of hydrophobic $Cu(OH)_2$ nanorods topped with hydrophilic $BaSO_4$ nanoparticles (Fig. 1a) via top interfacial growing (TIG) process (details of the preparation method see Supplementary Fig. S1a). The as-prepared Janus

substrate was denoted as Cu-SHBL. Additionally, drawing inspiration from the water-slippery nature of the Nepenthes pitcher plant, we constructed SLIPS on the Cu-SHBL substrate via underwater lubricating oil infusion, which could preserve the exposed hydrophilic $BaSO_4$ bumps (Fig. 1b, details of the preparation method see Supplementary Fig. S1b). For comparison, oil infusion was also conducted in air on the same substrate and served as a reference. Due to the superhydrophobic property of embedded fluorinated $Cu(OH)_2$ nanorods and the presence of hydrophilic $BaSO_4$ nanoparticles on top, a strong wetting gradient was formed. Therefore, during underwater oil infusion, water effectively encapsulated the hydrophilic $BaSO_4$ nanoparticles, allowing the lubricating oil to compactly and uniformly coat the substrate without contaminating the hydrophilic bumps and completely displacing air within the gaps of superhydrophobic nanorods, forming the Cu-SHBL-SLP film (Fig. 1b). In contrast, oil-infusion in air not only contaminated the hydrophilic $BaSO_4$ nanoparticles but also left air gaps within the film. This was reflected as the change in the amount of infused lubricating oil, where the underwater infused Cu-SHBL film has 22% higher oil infusion amount compared to the air-infused counterpart (Fig. 2a), indicating that the underwater infusion strategy is more effective in excluding air within the film and more conducive to even distribution of the lubricant (Supplementary Fig. S2).

As a protective measure during the oil infusion process, the hydrophilic $BaSO_4$ bumps on top of the Cu-SHBL films were largely preserved and remained exposed, as observed in the different fog capture processes using Environmental Scanning Electron Microscopy (ESEM), shown in Supplementary Fig. S3 (details of the process see Supplementary Movies S1 and 2). Furthermore, the Cu-SHBL film infused underwater exhibited a smaller water contact angle but a larger water sliding angle compared to its air-infused counterpart (Fig. 2b, c), further confirming the exposure of the on-top hydrophilic $BaSO_4$ nanoparticles. Typically, the loss of lubricating oil is the major

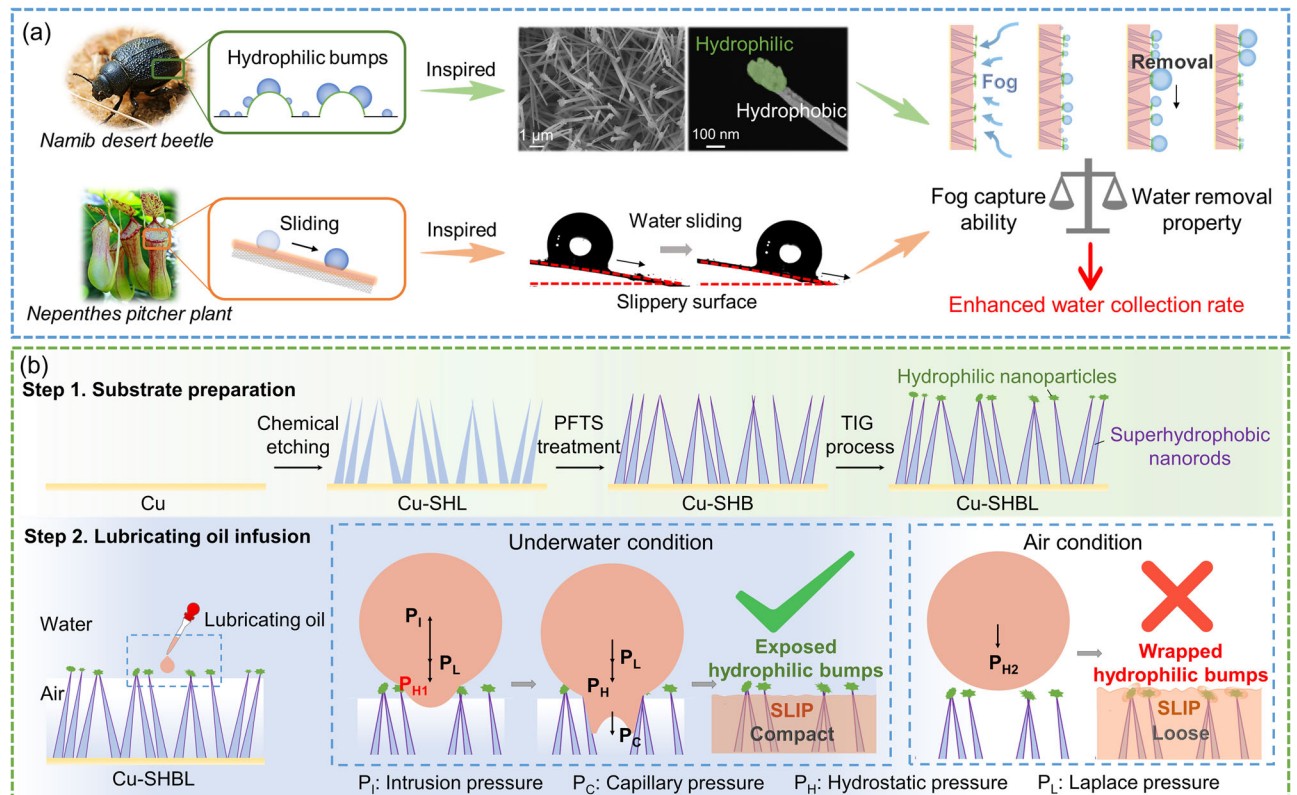

**Fig. 1 | The design of multi-bio-inspired SLIPS (slippery liquid-infused porous surfaces). a** Illustration of combining the Janus structure inspired by the desert beetle's back carapace with the SLIPS inspired by the Nepenthes pitcher plant for fog collection. **b** Substrate preparation and lubricating oil infusion steps for SLIPS with exposed hydrophilic bumps.

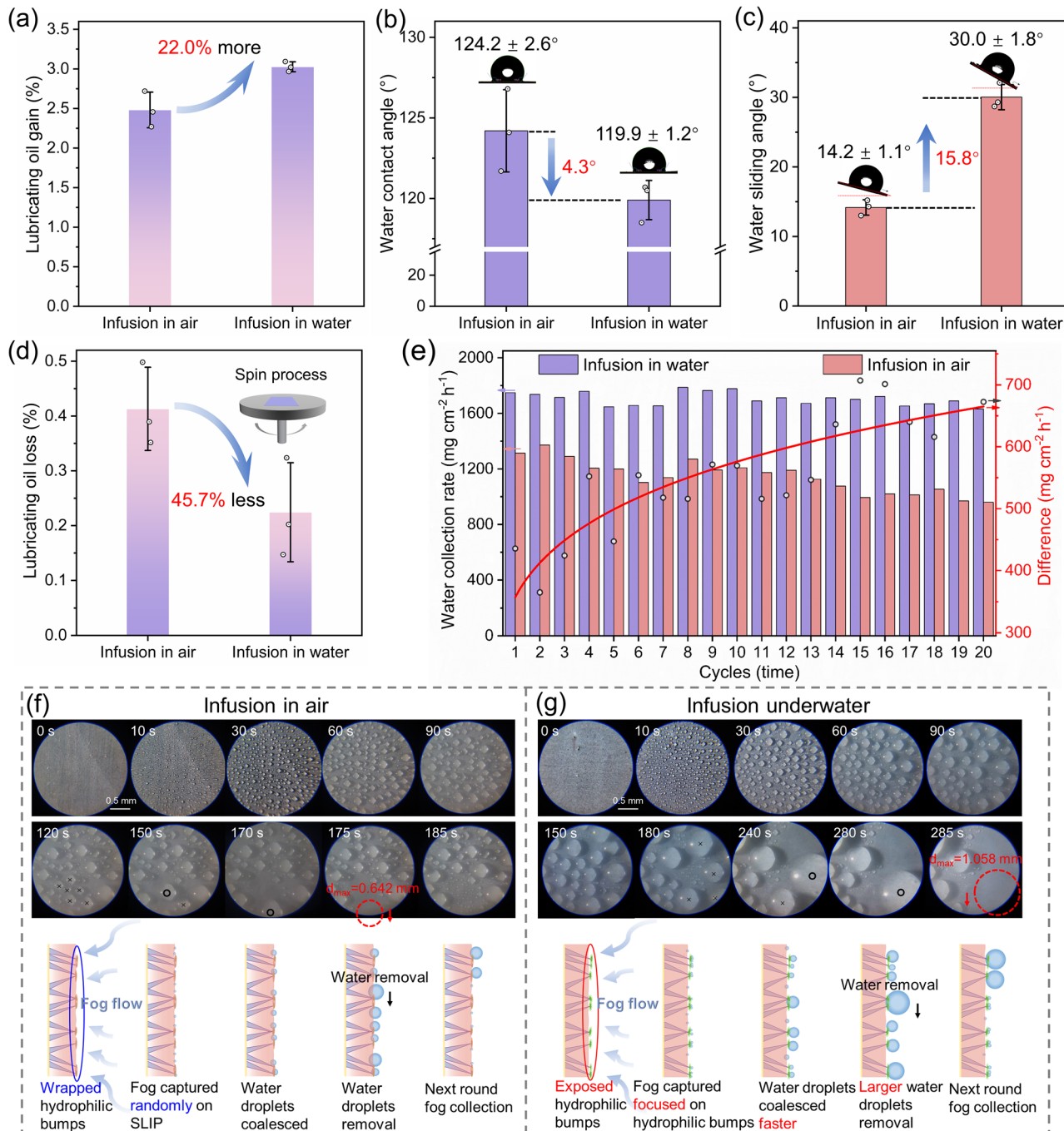

**Fig. 2 | Characterizations and fog collection performances on underwater-/air-infused Cu-SHBL-SLP (hydrophilic bumps on superhydrophobic arrays with slippery surface). a** The lubricating oil gains mass, **b** water contact angle, **c** water sliding angle, and **d** lubricating oil loss mass after the spin process of 1500 rpm for 60 s on underwater-/air-infused Cu-SHBL-SLP. Error bars in SD (standard deviation) are obtained by statistically repeating independent identical experimental results three times, and data are presented as mean ± SD. **e** The long-term fog collection performances and stabilities of underwater-/air-infused Cu-SHBL-SLP. (fog flow at 300 mL/h and every 30 min recorded as one cycle). The bars represent the water collection rate and correspond to the left Y-axis. The data points represent the value difference of the two bars (distinguished by color) and correspond to the right Y-axis. The red traces a trend line of the data points (difference), corresponding to the right Y-axis, and the fitting parameter is y = 357.26*x^0.21 ($R^2$ = 0.73). Online optical observations of the water collection process on **f** air-infused and **g** underwater-infused Cu-SHBL-SLP (fog flow at 100 mL/h). Source data are provided as a Source Data file.

limitation for long-term continuous applications of SLIPS[21]. Our underwater-infused SLIPS retained about 45.7% more lubricant than its air-infused counterpart after undergoing a spin process at 1500 rpm for 60 s (Fig. 2d), which also demonstrated commendable stability during long-term fog collecting applications for over 20 runs (Fig. 2e). The underwater-infused Cu-SHBL-SLP with exposed hydrophilic bumps demonstrated 33.2% higher water collection rate (WCR)

compared to its air-infused counterpart under the same experimental condition. This design featured exposed hydrophilic bumps on the hydrophobic SLIPS, which allowed the droplets to grow quickly and facilitated their removal. As illustrated in Figs. 2f, g, the underwater-infused Cu-SHBL-SLP showed a significantly larger critical droplet removal size of ~1.116 mm, in contrast to the ~0.642 mm observed with the air-infused version. Although the time to remove individual

droplets had increased, this led to the accumulation of larger droplets over the same period, enhancing the overall water collection rate.

## Surface wettability and water adhesion control

Furthermore, we compared the WCR of various film stages—Cu, Cu-SHL, Cu-SHB, Cu-SHBL, and Cu-SHBL-SLP (details of the various of films see Supplementary Fig. S4–S17) —to confirm the effectiveness of our design, which incorporated exposed hydrophilic bumps on SLIPS for collecting fog (Fig. 3a). As depicted in Fig. 3a and Supplementary Table S1, we systematically analyzed the fog collection performance of various surfaces with distinct wettability characteristics. The pristine Cu film, with its intrinsic hydrophobicity, exhibited a water collection rate (WCR) of 896.8 mg cm$^{-2}$ h$^{-1}$ and a water collection efficiency ($\eta$) of 8.45% (detailed information as seen in Methods section); The Cu-SHL film demonstrated the lowest performance (WCR: 844.0 mg cm$^{-2}$ h$^{-1}$, $\eta$: 7.95%), which can be attributed to its superhydrophilic surface. This extreme wettability leads to the formation of a dense water film, significantly impeding fog capture and droplet removal. The Cu-SHB film, with superhydrophobic modification, exhibited a higher performance (WCR: 1187.9 mg cm$^{-2}$ h$^{-1}$, $\eta$: 11.20%) due to its water-repellent properties and improved droplet transport dynamics. The integration of hydrophilic components in the hybrid Cu-SHBL film further improved the WCR to 1317.8 mg cm$^{-2}$ h$^{-1}$ and $\eta$ to 12.40%, as the hydrophilic regions enhanced fog capture capability. The Cu-SHBL-SLP film demonstrated the best performance (WCR: 1663.7 mg cm$^{-2}$ h$^{-1}$, $\eta$: 15.68%). This significant enhancement is attributed to the presence of exposed hydrophilic bumps combined with the ultra-low sliding angle and minimal frictional resistance of the bioinspired slippery liquid-infused porous surface (SLIPS), which synergistically optimizes both fog capture and droplet removal. Generally, the optimal scenario for fog-collection was featured by a shorter time and a heavier weight for the first droplet removal (FDR). We used a high-speed camera to measure the FDR time and weight across different films (Fig. 3b). Results showed that on the hydrophilic surfaces of Cu and Cu-SHL, although the FDR weight was very large (>0.3 g), the FDR time was significantly long (>350 s) due to strong water adhesion. Following the hydrophobic treatment reduced surface adhesion to water, the Cu-SHB and Cu-SHBL films exhibited significantly shorter FDR times. However, the absence of water aggregation centers on a super-hydrophobic surface resulted in small first droplets, with the FDR weight less than 10$^{-4}$ g (These tiny droplets, too small to weigh, were estimated using magnified images during the water collection process, as detailed in the Supplementary Figs. S15 and S16). The introduction of underwater oil infusion on the Cu-SHBL-SLP film resulted in both a shorter FDR time and an FDR weight greater than 0.01 g. This confirmed that the hydrophilic bumps on SLIPS acted as effective central sites for tiny water droplets capturing and coalescing. Detailed observations of the corresponding water collection process were available in the Supplementary Fig. S13-S17.

By adjusting the amount of BaSO$_4$ hydrophilic bumps on Cu-SHBL-SLP (details of the samples morphology see Supplementary Fig. S18), we identified the optimal hydrophilic bump loading at a Ba content of 0.53 $_{wt}$% (Fig. 3c and Supplementary Table S3). Notably, before oil infusion, an increase in the number of BaSO$_4$ hydrophilic bumps led to greater adhesion of the Cu-SHBL to water droplets, which consequently increased the rolling angle of individual droplets (Supplementary Figs. S19 and S20). This trend of enhanced water adhesion not only persisted but intensified after underwater oil infusion, confirming that our strategy effectively preserved the exposed hydrophilic bumps (Fig. 3d, e). Note that an excessive number of hydrophilic BaSO$_4$ bumps can even hinder the surface transport of water droplets, leading to a reduced WCR (Fig. 3c). Using a high-speed camera, we observed the average droplet removal size on Cu-SHBL-SLP surfaces (positioned at 90 degrees) with varying numbers of BaSO$_4$ hydrophilic bumps and calculated corresponding water adhesion constants

(Supplementary Fig. S21–24). Result showed that the adhesion constant of droplets to the film surface increases exponentially with the number of hydrophilic bumps, supporting our conclusions (Fig. 3f, Supplementary Fig. S23 and S24). A comparison of Cu-SHBL-SLP films with different contents of hydrophilic bumps revealed that the sample with a 0.53 $_{wt}$% Ba content offers the optimal balance, resulting in the optimal FDR time and weight (Fig. 3g).

## Initial capture of water droplets from microscopic views

Exposed hydrophilic bumps enhanced adhesion to water molecules, which accelerated the capture and coalescence of water droplets. To substantiate this observation at the microscopic level, we monitored the dissociation of water molecules adsorbed on Cu-SHBL-SLP surfaces with varying hydrophilic bump contents (Ba content of 0 - 1.54 $_{wt}$%) using in situ diffuse reflection infrared Fourier transform spectroscopy (DRIFTS). Under continuous water vapor flow, the DRIFTS spectra from different Cu-SHBL-SLP surfaces showed three types of infrared absorption bands located at 3447 cm$^{-1}$, 3250 cm$^{-1}$, and 1644 cm$^{-1}$. These bands correspond to the dissociated water's O-H stretching vibrations, water molecules' O-H stretching vibrations, and H-O-H bending vibration, respectively (Fig. 4a). The results demonstrated that as the number of hydrophilic bumps increased, the efficiency of water molecule dissociation significantly improved (Fig. 4b, c). This revealed that BaSO$_4$ hydrophilic bumps on SLIPS promoted the chemical dissociation equilibrium of water molecules, thereby increased water affinity and accelerated the formation of initial droplets and subsequent aggregation of fog droplets (Fig. 4d, e).

## From 2D patterns to 3D arrays

Leveraging the characteristic of exposed hydrophilic bumps on SLIPS, we further developed reverse Christmas tree-shaped Cu-SHBL-SLP patterns, inspired by desert cacti (Fig. 5a). Through laser ablation, we precisely adjusted the length (L) and inclination angle ($\theta$) of the serrated edges of the pattern templates (Supplementary Fig. S25 and S26). This adjustment imparted unique Laplace pressures to the 2D sheet that influence the coalescence and transportation of water droplets[22] (Fig. 5d, Supplementary Movie S3). This distinctive patterned structure facilitated the coalescence and transportation of multiple droplets within ~72 s, which was four times faster than the pristine, unpatterned Cu-SHBL-SLP film (>285 s). In this structure, the length L should not be overly long. An appropriate L provided sufficient Laplace pressure to aid water movement, but an excessively high L increased the transportation distance of the droplets from the tip to the base, thereby adding unnecessary time for droplet coalescence in the central channel (Fig. 5b and Supplementary Table S4). The inclination angle $\theta$ was also important; generally, the smaller the $\theta$, the greater the combined effect of gravity and Laplace forces, which enhanced the efficiency of droplet coalescence and transportation. When $\theta$ exceeded 90 degrees, gravity counteracted the Laplace pressure. However, within a range of $\theta$ = 30 - 60 degrees, an increase in $\theta$ benefited the initial fog capture process, making $\theta$ = 60 degrees the optimal parameter for WCR (Fig. 5c and Supplementary Table S5). This optimization is based on the observation that a smaller inclination angle impeded the initial fog capture along the span direction, negatively affected the WCR (Supplementary Fig. S27 and Movie S4). Under the conditions of L = 1 cm and $\theta$ = 60 degrees, the patterned Cu-SHBL-SLP achieved the WCR over 5000 mg/cm² h at 300 mL/h and reached up to 60000 mg/cm² h at 1500 mL/h, marked the highest water collection performance reported to date with fog flow outputs ranging from 420 to 1500 mL/h (Fig. 5e). Note that water collection performance was largely depended on the water content in the ambient fog environment. Our analysis only compared the top 20 representative fog collection systems with clearly specified fog flow conditions and no external energy input[16,23–40] (Fig. 5e and Supplementary Table S6), and

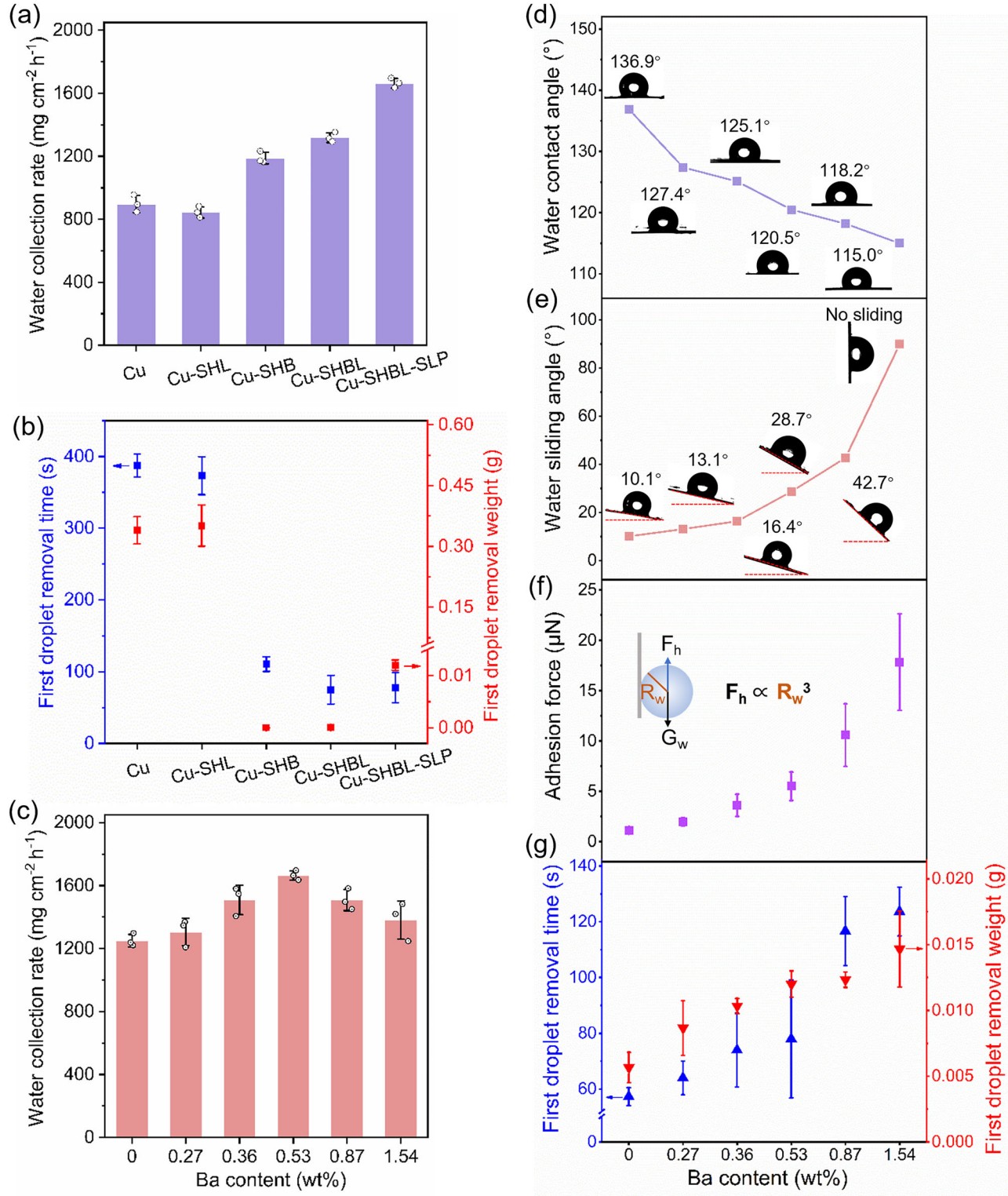

**Fig. 3 | Surface wettability and water adhesion control over various-stage films.** **a** Water collection rate (WCR) (fog flow at 300 mL/h) and **b** the comparison of first droplet removal (FDR) time and weight (fog flow at 100 mL/h) of the various samples (original Cu, Cu-SHB, Cu-SHL, Cu-SHBL-0.53% and Cu-SHBL-0.53%-SLP films). **c** Comparisons of WCR (fog flow at 300 mL/h), **d** water contact angle, **e** water sliding angle, **f** calculated droplet adhesion force, **g** FDR time and weight (fog flow at 100 mL/h) of various Cu-SHBL-SLP with different Ba content (Ba content was quantified by using EDS analysis as seen in Supplementary Table S2). Error bars in SD (standard deviation) are obtained by statistically repeating independent identical experimental and results three times, and data are presented as mean ± SD. Source data are provided as a Source Data file.

our system surpassed all previously reported systems when the output fog flow exceeded 420 mL/h.

To provide a more objective assessment of the water collection performance and ensure fair comparisons with literature results, we have introduced the water collection efficiency (η) as a key evaluation metric. This efficiency is defined as the ratio of the experimental water collection rate ($W_{exp}$) to the theoretical water collection rate ($W_{the}$)[41]. For example, the 2D-patterned Cu-SHBL-SLP film achieves a water

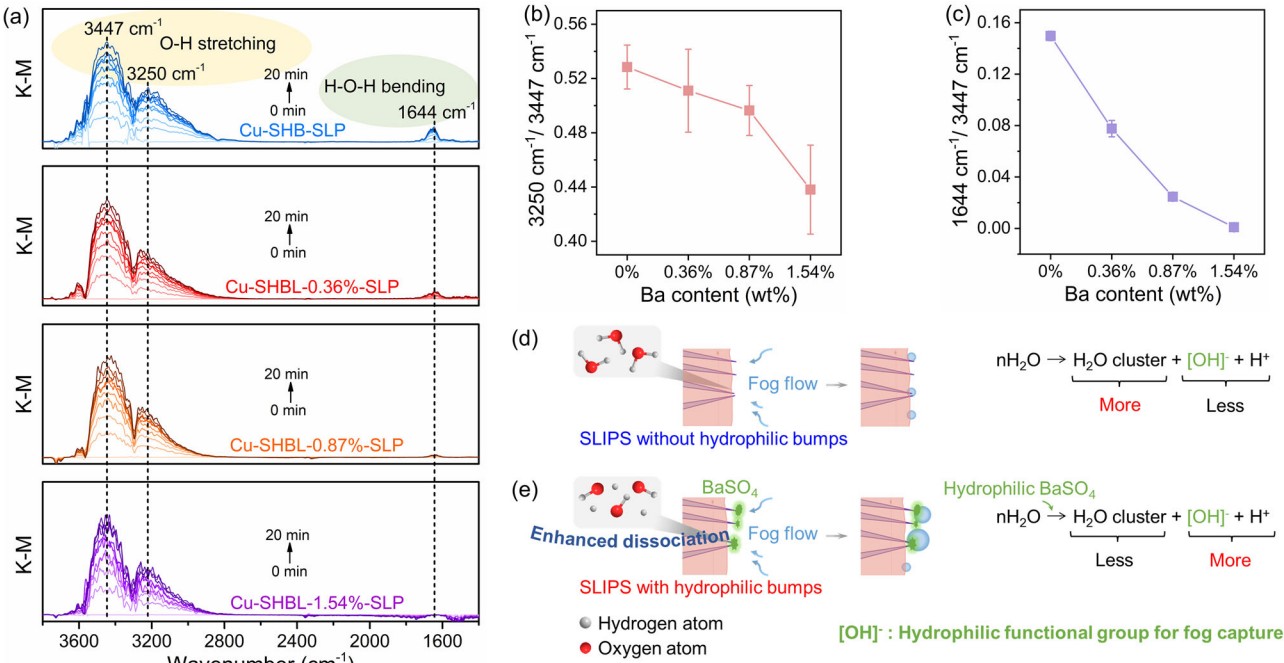

**Fig. 4 | Water dissociation and capture mechanism on SLIPS with/without on-top hydrophilic bumps. a** In situ DRIFTS (diffuse reflection infrared Fourier transform spectroscopy) spectra collected from the continuous $H_2O$ adsorption on various Cu-SHBL-SLP film surfaces with different Ba contents. **b** Peak ratio (y-axis) of the DRIFTS intensity at 3250 $cm^{-1}$ to 3447 $cm^{-1}$ and (**c**) 1644 $cm^{-1}$ to 3447 $cm^{-1}$ of various Cu-SHBL-SLP films. Error bars in SD (standard deviation) are obtained by statistically repeating independent identical experimental results three times, and data are presented as mean ± SD. **d, e** Illustrations show the surface dissociation state of water when the SLIPS with/without hydrophilic bumps are contacted to the fog flow. Source data are provided as a Source Data file.

collection rate (WCR) of 5081.25 mg $cm^{-2}$ $h^{-1}$ and a water collection efficiency (η) of 47.89% (Supplementary Table S7). This significant improvement is primarily attributed to the unique water transport mechanism on the surface, driven by Laplace pressure, which enhances both fog capture and droplet removal. Additionally, a single 2D-patterned Cu-SHBL-SLP collector (active area ≈ 0.646 $cm^2$) costs approximately 3.88 yuan ($0.54) and achieves a water collection rate of 61588 mg $cm^{-2}$ $h^{-1}$ (fog flow at 1500 mL/h). Consequently, this corresponds to a remarkably low water production cost of ≈0.006 yuan ($0.0008) per liter per square meter per hour (fog flow at 1500 mL/h), performed superior cost efficiency and scalability (Supplementary Table S8). Furthermore, inspired by the fog collection enhancement effect observed in densely arrayed plants in the forest[42], we constructed an integrated fog collection device by assembling a 2D patterned sheet into densely arranged 3D arrays to maximize the extraction of moisture from the air (Fig. 5f). The sealed box beneath the arrays served as storage to minimize loss from volatilization. The fog collection device consistently collected more than 660 g of water within 500 min (Fig. 5g, h).

Moreover, considering the dual challenges of water scarcity and pollution, integrating water collection with purification processes addresses two critical environmental issues simultaneously. This approach is particularly vital in regions affected by both limited water resources and high levels of atmospheric pollutants, such as the volatile organic compounds (VOCs), environmental pesticide residues, and other organic pollutants, can lead to the contamination of collected water, rendering the limitation of direct use[43,44]. Traditional methods of water purification often require extensive infrastructure and significant energy inputs, which are not always feasible in remote or economically disadvantaged areas. By developing systems that can both collect and purify fog, we can harness naturally occurring water sources and ensure that the water is safe for consumption and other uses immediately upon collection[45]. Leveraging the unique structure of hydrophilic bumps on SLIPS, hydrophilic photocatalysts, i.e., $TiO_2$

nanoparticles, widely used in photocatalytic water pollution remediation[46], can easily adhere to the hydrophilic bumps due to similar wettability, thereby endowing these exposed tips with photocatalytic water purification capabilities (Fig. 5i, j). By placing as-prepared Cu-SHBL(Ti)-SLP film under UV light illumination, tiny fog droplets are purified by a significant amount of reactive oxygen species (ROS) generated during the initial capture process by the $TiO_2$ nanoparticles. These tiny initial droplets also act as microreactors with a high gas-liquid exchange capacity and enhanced local ROS concentration; thus, their purification effect is usually more efficient than in bulk water[47]. We investigated the photocatalytic purification capacity of the $TiO_2$-topped SLIPS sheet by adding the representative organic dye (Rhodamine B) pollution to the fog sources to simulate the photodegradation of organic pollutants during collection under UV irradiation, and the results showed that the dye molecules in the collected water were completely degraded without any residue (verified by both the photo images and UV-vis spectrophotometer) (Fig. 5k, l). More importantly, during this process, the water collection performance of the SLIPS films was not compromised (Supplementary Fig. S28), confirming the high efficiency of this design in both water collection and purification.

## Discussion

This work of ours, from the microscopic to the macroscopic, developed a comprehensive solution to water sustainability by combining multiple bioinspired designs (desert beetle back, pitcher plant slippery surface, cactus patterned structure, and densely arrayed plants) with advanced material science. By maintaining the structure of hydrophilic bumps on SLIPS and integrating $TiO_2$ photocatalysts, our system efficiently collects and purifies fog, gathering over 660 grams of water in 500 min. The key to this design is the successful preservation of the exposed hydrophilic bumps through the underwater lubricant-oil infusion process, which resolves the longstanding dilemma of the water capture vs. transportation balance in the field. This

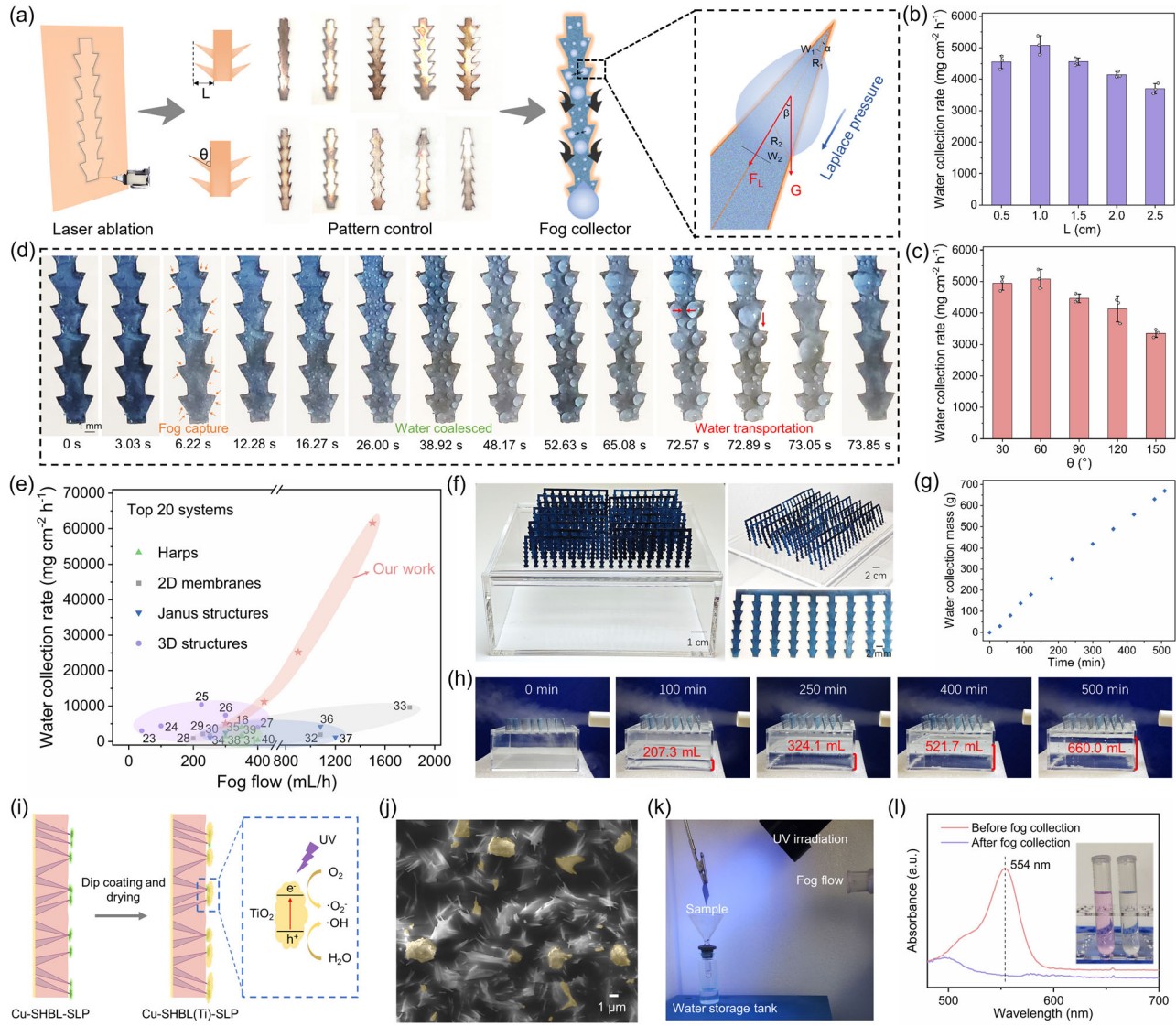

**Fig. 5 | Macro-design of the 3D fog collector and device. a** The 2D patterned fog collector fabrication process. Laser ablation technology could control the L (length) and θ (inclination angle) of the template serrated edge. Transportation of water droplet is driven by the combined effect of G(gravity) and F_L (Laplace pressure), and F_L is subjected to R (curvature radius), w(half-width) and α (semiapex angle) of the conical structure. β represents the angle between G and F_L. **b** WCR (water collection rate) of the 2D patterned fog collector with different L and (**c**) θ values. Error bars in SD (standard deviation) are obtained by statistically repeating independent identical experimental results three times, and data are presented as mean ± SD. **d** The actual fog collection process of the optimal 2D patterned fog collector (fog flow at 100 mL/h) captured by high-speed camara; **e** comparison of WCR between our work and other representative works (details of the references and experimental parameters were summarized Supplementary Table S6); **f** assembled 3D fog collection device and **g** its water collection performance (fog flow at 1500 mL/h); **h** the 3D fog collector could collect over 660 mL water within 500 min, and the sealed box was filled with collected water; **i** the preparation process of Cu-SHBL(Ti)-SLP sheet and **j** the corresponding SEM image. The SEM image is representative of three independent analysis that yielded similar result. **k** The lab-scale set-up of the water collection system synergistically integrated with organic pollutant degradation and (**l**) the results of the Rhodamine B degradation during fog collection (fog flow at 300 mL/h). Source data are provided as a Source Data file.

breakthrough ensures strong water capture capabilities along with efficient transportation, thus achieving record-breaking water collection results under mid-to-high fog flow conditions (420-1500 mL/h). The future perspective of this work aims to scale and refine this technology, enhancing its application potential through the development of portable, energy-efficient, and cost-effective products.

## Methods

### Chemicals

Copper sheet (0.1 mm) was purchased from Biling factory. 1H,1H,2H,2H-Perfluorooctyltriethoxysilane (PFTS) was obtained from Bidepharm. Sodium hydroxide (NaOH), ammonium persulfate ((NH$_4$)$_2$S$_2$O$_8$), barium chloride (BaCl$_2$), potassium sulfate (K$_2$SO$_4$),

ethanol, and acetone were purchased from Sinopharm Chemical Reagent Co., Ltd. Polyperfluoromethylisopropyl ether (PFPE) was gained from Aladdin Reagent Co. Deionized water (DI water) was used in all the experimental processes. All the chemicals were applied without any other purification.

### Preparation of Cu-SHBL film

The original copper sheet (20 mm × 20 mm) was first washed with detergent, acetone, ethanol, and DI water, respectively. Then the cleaned Cu sheet was immersed in the mixed solution containing 2.5 m NaOH and 0.13 m (NH$_4$)$_2$S$_2$O$_8$ for 15 min to obtain the Cu/Cu(OH)$_2$, named as Cu-SHL. To endow the water-repellent ability, the Cu-SHL was immersed in the PFTS ethanol solution and dried at 50 °C in air,

which was named as Cu-SHB. Then, the superhydrophobic nanorods with hydrophilic particles on their top were achieved via a unique top interfacial growth (TIG) process. Firstly, the superhydrophobic Cu-SHB was clamped by two glass tubes with a diameter of 1.5 cm. Then, 0.1 m $BaCl_2$ aqueous solution was poured into the glass tube above. The aqueous solution could not wet the beneath sheet and kept a Cassie state due to the low surface energy and high roughness of the fluorinated $Cu(OH)_2$ nanorod arrays. The superhydrophobic nanorods could prevent the sheet from being wetted while only the top of the nanorods could contact the aqueous solution, which provided a feasible strategy to realize the top modification. A dropper with 0.1 m $K_2SO_4$ aqueous solution was forced into the above aqueous solution and ejected onto the top surface of Cu-SHB. Consequently, the $BaSO_4$ sediment could be formed on the nanorods and regarded as the Cu-SHBL film.

## Underwater lubricating oil infusion process

To construct the slippery liquid-infused porous surface (SLIPS), PFPE was selected as the lubricating oil and the underwater oil infusion process. Firstly, the prepared Cu-SHBL film was placed at the bottom of a Petri dish filled with DI water under external force. Then, 500 μL PFPE was taken by a pipette gun and dropped onto the Cu-SHBL film surface, realizing the underwater lubricating oil infusion process. After being placed in water for 10 minutes, the sheet was vertically dried at room temperature overnight. And the obtained sheet was named as Cu-SHBL-SLP. To make a comparison, the Cu-SHB was used as the substrate and conducted the underwater lubricating oil infusion process was conducted, which was named Cu-SHB-SLP.

## Water harvesting measurement

The water harvesting process of the prepared material was evaluated by the homemade testing system. The fog flow with 300-1500 mL/h was provided by the ultrasonic humidifier[48] (YADU ultrasonic humidifier YC−D205, China and AUX mechanical standards, Hefei Hemei E-commerce Co., Ltd.). The temperature and relative humidity (RH) were 20 °C and 90%, respectively. All the studied meshes were clamped on a holder with a fixed distance of 6 cm to the humidifier nozzle, and all the valid collection areas of the meshes were 3.14 cm² for 2D sheet and a different valid area for the 3D patterned sheet. Without any further illustration, the fog flow was normally perpendicular to the mesh surface. The collected water was stored using a self-made container, and the mass of the collected water was calculated every 10 minutes. The water collection mass (WCM) was calculated with the following equation:

$$WCM = \frac{M}{A} \tag{1}$$

where M (mg) is the amount of the collected water and A (cm²) is the valid area of the meshes. The water collection rate (WCR) was calculated with the following equation:

$$WCR = \frac{WCM}{t} = \frac{M}{A \times t} \tag{2}$$

where t (h) is the experimental time of fog collection.

## Water collection efficiency

The water collection' efficiency is defined as the ratio of the experimental water collection rate ($W_{exp}$) to the theoretical water collection rate ($W_{the}$), expressed as follows[41]:

$$\eta = \frac{W_{exp}}{W_{the}} \times 100\% \tag{3}$$

where $W_{exp}$ and $W_{the}$ are the measured experimental water collection rate and calculated theoretical water collection rate of the samples. The effect of experimental and setup conditions could be eliminated. The calculation of maximum available fog for collection follows the turbulent jet theory presented by Cushman-Roisin[49]. In this work, fog flow with 300 mL/h was outputted by a nozzle with the diameter of 2.2 cm, and the Reynolds number was calculated as 15,181, which was termed as turbulent flow and suitable for applying the general theory. The theoretical water collection rate ($W_{the}$) could be calculated as follows:

$$W_{the} = u_{max} \times c_{max} \tag{4}$$

where $u_{max}$ and $c_{max}$ were the fog velocity and fog concentration at the sample surface. As the fog jet propagates downstream from the nozzle, its velocity and absolute humidity exhibit progressive decay, as follows:

$$u(x, r) = \int u_{max}(x) e^{-50r^2/x^2} dr \tag{5}$$

$$c(x, r) = \int c_{max}(x) e^{-50r^2/x^2} dr \tag{6}$$

Velocity and humidity concentration exhibit non-uniform distributions across the sample surface, with both parameters following Gaussian-like profiles that peak centrally, as illustrated below:

$$u_{max}(x) = u_0 \times \frac{5d}{x} \tag{7}$$

$$c_{max}(x) = c_0 \times \frac{5d}{x} \tag{8}$$

where $u_0$ and $c_o$ are the velocity and concentration of the nozzle jet, and the variable x quantifies the distance between the sample surface and the jet cone apex. In this work, x value of the low-powered ultrasonic humidifier (YADU) with the fog flow at 300 mL/h was around 30 cm, and the high-powered ultrasonic humidifier (AUX) with the fog flow of 420−1500 mL/h was around 20 cm. Meanwhile, the air was saturated with moisture at 20 °C, and the $c_o$ value was 17 g/m³. The velocity could be calculated as follows:

$$u_0 = \frac{Q}{S} \tag{9}$$

where S was the nozzle area. In this work, the nozzle diameter d = 2.2 cm, and the S was calculated as 3.8 cm². When the low-powered ultrasonic humidifier was applied, the carrier gas flow rate (Q) could be calculated as 294.2 L/min, and the $u_0$ could be calculated as 12.90 m/s. Therefore, the $W_{the}$ could be calculated as 10610.3 mg cm⁻² h⁻¹, and all the water collection efficiency (η) could be calculated according to the measured experimental water collection rate ($W_{exp}$).

## Lubricating oil gain/loss tests

The lubricating oil gain was determined by the weight of the samples before and after the lubricating oil infusion process. The lubricating oil gain could be calculated as follows:

$$Lubricating\ oil\ gain(\%) = \frac{M_{gb} - M_{ga}}{M_{gb}} \tag{10}$$

where $M_{gb}$ (g) was the weight of the material before lubricating oil, $M_{ga}$ (g) was the weight of the material after lubricating oil. Due to the superoleophilic property of superhydrophobic Cu-SHB, the lubricating oil gain test was conducted on two sides.

The lubricating oil loss was determined by the weight of samples before and after the spin process or the fog collection test. The lubricating oil loss could be calculated as follows:

$$\text{Lubricating oil loss}(\%) = \frac{M_{lb} - M_{la}}{M_{lb}} \qquad (11)$$

where $M_{lb}$ (g) was the weight of the material before tests, $M_{la}$ (g) was the weight of the material after tests.

## Characterization devices

The scanning electron microscopy (SEM, JSM-7800F, Hitachi, Japan) equipped with the energy-dispersive spectroscope (EDS) was used to study the surface morphology and structure. The environmental scanning electron microscope (ESEM, QUANTA200) was applied to study the water capture process of the prepared sample's surface. The UV-visible spectrophotometer (Agilent Cary 8454) was used to test the organic pollution contents. The surface wettability was observed by the contact angle meter (LAUDA Scientific, China). The water droplet adhesive force of the sample surface was measured via the Surface Tension Measuring Instrument (Kruss K100, German). Fourier transform infrared spectrometer (Nicolet Nexus 470, America), X-ray diffraction (XRD-6100, Japan) and X-ray photoelectron spectroscopy (XPS, Escalab 250Xi, America) were tested to analyze the chemical components and surface functional groups of the sample. In-situ diffuse reflection infrared Fourier transform spectroscopy (DRIFTS) experiments were performed on a Thermo Nicolet iS10 spectrometer equipped with a mercury cadmium telluride (MCT) detector to study the dissociation of water molecules adsorbed on the prepared sample's surface during the fog collection process.

## Reporting summary

Further information on research design is available in the Nature Portfolio Reporting Summary linked to this article.

## Data availability

The data generated in this study are provided in the Source Data file. Source data are provided with this paper.

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

## Acknowledgements

This work was financially supported by the National Natural Science Foundation of China (Nos. 22378173) (J Dai), Senior Talent Research Foundation of Jiangsu University (No. 22JD017) (Y. Yan), and Graduate Research Innovation Program of Jiangsu Provincial (KYCX23_3648) (J. Wu).

## Author contributions

J. Wu: conceptualization, methodology, investigation, writing—original draft. C. Li: investigation, methodology, supervision. J. Dai: conceptualization, methodology, supervision, funding acquisition. Y. Yan: conceptualization, supervision, writing—review and editing, funding acquisition.

## Competing interests

The authors declare no competing interests.
