## [Transparent Peer Review file · Nature Communications]

Preserving exposed hydrophilic bumps on multi-bioinspired slippery surface arrays unlocks high-efficiency fog collection and photocatalytic cleaning

Corresponding Author: Professor Yan Yan

Version 0:

Reviewer comments:

Reviewer #1

(Remarks to the Author)

This study presents a groundbreaking approach to fog water collection by ingeniously integrating hydrophilic bumps on SLIPS, effectively resolving the long-standing trade-off between nucleation and transport efficiency. The novel underwater lubricating oil infusion strategy ensures the stable preservation of hydrophilic sites, significantly enhancing water collection performance. The seamless combination of multiple bioinspired principles—desert beetle structures, pitcher plant SLIPS, and cactus-patterned designs—demonstrates an exceptional level of innovation and interdisciplinary insight. If the authors can provide answers to the following questions, I am in favor of the publication of this study in the journal.

- How does the bioinspired SLIPS design proposed in this study fundamentally improve upon existing fog collection technologies that utilize hydrophilic/hydrophobic patterning?
- The authors should further discuss the cost efficiency and scalability of this technology for large-scale practical applications.

Reviewer #2

(Remarks to the Author)

In this work the authors present a new method for the fabrication of biphilic SLIP surfaces using top interfacial growing of BaSO₄ nanoparticles to develop hydrophilic tips on Cu(OH)₂ nanorods, followed by a very smart underwater lubricant infusion of the substrate, which allows the preservation of the hydrophilicity of the tips. This is a new part in terms of fabrication. Another smart and new element was the fabrication of 3D channels on such surfaces, which was also presented, followed by the fabrication of a prototype device. The samples were used for the application of fog collection, providing high collection rates, although comparisons with the literature cannot be conclusive as I will explain below.

In general, the paper is very interesting with significant novelty and the characterization is detailed and accompanied by simulations, although this is a part that I am not an expert to discuss. I am satisfied with the fabrication and characterization. However, I find that there are some misconceptions and some potentially not fully justified comparisons with the literature, which if not clarified significantly undermine the value of this work in any journal that it may be published. Below you can find comments that can improve the clarity of the manuscript and address these points.

1) The authors mention that the interplay between nucleation and transportation is the key factor for fog collection. However the authors use fog collection.

There is a common misconception at the literature of atmospheric water harvesting between the two collection methods, fog and dew collection. During fog collection, a mixture of air and water droplets come in contact with the collection device and the water is being captured due to the adhesion of it with the surface, in that case there is no nucleation or condensation, as there is no phase change of water vapor to liquid vapor (as in the case of dew). To that extent, the authors should revise their manuscript to correct this narrative. Therefore, I would suggest to study more carefully related works such as those below to understand the differences and avoid any confusion or misinterpretation. This is a comment related to the introduction not the experimental work, and thus it does not affect the validity of the results presented, but it should be clearly addressed in the introduction.

Lu, H.; Shi, W.; Guo, Y.; Guan, W.; Lei, C.; Yu, G. Materials Engineering for Atmospheric Water Harvesting: Progress and Perspectives. *Adv. Mater.* 2022, 34 (12), 1–16. <https://doi.org/10.1002/adma.202110079>.

That you have already included in your references, as well as

Nioras, D.; Ellinas, K.; Gogolides, E. Atmospheric Water Harvesting on Micro-Nanotextured Biphilic Surfaces. *ACS Appl. Nano Mater.* 5 (8), 11334–11341. <https://doi.org/10.1021/acsanm.2c02439>.

and

Seo, D.; Lee, J.; Lee, C.; Nam, Y. The Effects of Surface Wettability on the Fog and Dew Moisture Harvesting Performance on Tubular Surfaces. *Sci. Rep.* 2016, 6 (March), 1–11. <https://doi.org/10.1038/srep24276>.

2) Fog collection results at figure 2a) are the most important to explain fog collection mechanism on the newly introduced surfaces compared to uniform wettability ones, thus they should be analysed in detail in text as well, explaining the reason of the differences between the surfaces.

3) In Figure 4 e) and in supplementary data the authors make an attempt to compare the results of the literature with this work on WCR, based on the fog flow in each experiment. Although this graph is a good indication for the general state of the fog collection, and the rather good results of this work, there are many more factors that contribute to the WCR, such as the distance from the fog source, the relative sample size to the fog flow projection e.t.c. Therefore it may be that comparisons are not always easy or "fair". The following work has introduced an efficiency factor in order to allow fair comparison of literature results in different setups. Perhaps, using an efficiency factor such comparison may enhance even more or reduce the performance of the surfaces introduced in this paper.

Nioras, D.; Ellinas, K.; Constantoudis, V.; Gogolides, E. How Different Are Fog Collection and Dew Water Harvesting on Surfaces with Different Wetting Behaviors? *ACS Appl. Mater. & Interfaces* 2021, 13 (40), 48322–48332. <https://doi.org/10.1021/acsami.1c16609>.

Reviewer #3

(Remarks to the Author)

In their manuscript

"Preserving exposed hydrophilic bumps on multi-bioinspired slippery surface arrays unlocks high-efficiency fog collection and cleaning",

the authors describe the design, manufacturing and testing of a fog collection device. The function of the device combines a hydrophilic tip with

a slippery hydrophobic surface in order to allow faster drop collection by focusing drop growth to small hydrophilic zones.

I was asked to comment only on the MD simulations.

The actual device describes superhydrophobic nanorods infused in an oil film. The key facts amenable to MD simulation are droplet nucleation and growth.

One might also try to simulate slippage or the manufacturing process.

The MD results show a system made of PFPE and a BaSO₄ substrate. It should be noted that the actual nanorods are made of (functionalized) Cu(OH)₂ and are not modeled, however the main problem

is clearly the PFPE lubricating oil. Because the MD simulation is very short (0.5 ns), and the water concentration is very high, the PFPE does

not have the time to equilibrate into a proper hydrophobic film. Instead, we observe droplet formation inside the PFPE (3f), which should simply not happen.

Obviously, the system must be equilibrated before adding the water film. It might even be that the force field parameters predict hydrophilic PFPE, which would mean they are not adequate.

The ultra-short timescale at high water content and low temperature just shows that the instantaneously formed droplet attaches itself to the BaSO₄, in other words, that the BaSO₄ surface is indeed hydrophilic. There has been no serious attempt to study of nucleation rate of droplets and wetting behavior at appropriate conditions. If this was done, the simulation might be more convincing.

However, the core of the paper is not the discovery that BaSO₄ crystals will bind to water droplets or that PFPE is hydrophobic.

If anything it is about the "trick" to keep them exposed by performing the manufacturing process in water, i.e. keeping the BaSO₄ tip wet while oiling the nanorods. I think this process would be a more interesting thing to study. Alternatively,

continuum modelling could be performed, if surface tensions were known, of the droplet binding and slippage behavior. Furthermore, the computational details contain some trivial "equations" for LJ and Coulomb potentials, a reference to LAMMPS and mentioning materials studio. Instead of the equations, details on the potentials used should be given. To

make the simulation reproducible, the precise parameters used for water, PTFE and BaSO₄ should be stated (in the SI) or referenced. If new parameters were used, a validation procedure will have to be reported as well. The MD details also miss

references to treatment of boundary conditions (periodic electrostatics ? Barrier in z ?) and the correct references of the methods used. Referencing classical MD is not necessary and the reference is not to the first use of the method. These are

beginner's mistakes.

Despite the pretty images, I do not think these simulations add any scientific value at present and should be removed or completely reworked.

The recommendation "do not publish" pertains only to the MD data.

Version 1:

Reviewer comments:

Reviewer #1

(Remarks to the Author)

This study was conducted with many experiments and detailed analyses, and through meticulous commentary and supplementation, I agree to include it in this journal.

Reviewer #2

(Remarks to the Author)

The authors have adequately addressed the comments and questions raised in the first review, and therefore I recommend the publication of the article. One suggestion is that the water collection efficiency calculations could be moved to the Supplementary Information to make the main text more concise and easier to read.

Response to Reviewers

Reviewer 1:

Comment: This study presents a groundbreaking approach to fog water collection by ingeniously integrating hydrophilic bumps on SLIPS, effectively resolving the long-standing trade-off between nucleation and transport efficiency. The novel underwater lubricating oil infusion strategy ensures the stable preservation of hydrophilic sites, significantly enhancing water collection performance. The seamless combination of multiple bioinspired principles—desert beetle structures, pitcher plant SLIPS, and cactus-patterned designs—demonstrates an exceptional level of innovation and interdisciplinary insight. If the authors can provide answers to the following questions, I am in favor of the publication of this study in the journal.

Response: We sincerely appreciate your valuable time and positive comments on our manuscript. We have carefully revised the manuscript in response to all points raised. The revised manuscript has been submitted, and we believe the revisions have been incorporated comprehensively, resulting in a more readable manuscript.

Q1. How does the bioinspired SLIPS design proposed in this study fundamentally improve upon existing fog collection technologies that utilize hydrophilic/hydrophobic patterning?

Response: Thank you for your insightful review. The proposed slippery liquid-infused porous surface (SLIPS) design fundamentally enhances fog collection by significantly accelerating droplet transport, owing to its ultra-low water friction and minimal sliding angle compared to conventional hydrophobic surfaces. Importantly, the integration of exposed hydrophilic bumps within the SLIPS framework ensures robust and efficient fog capture. This innovative hydrophilic/SLIPS pattern effectively addresses the inherent trade-off between water capture and transport, enabling a remarkable improvement in water collection performance.

Q2. The authors should further discuss the cost efficiency and scalability of this technology for large-scale practical applications.

Response: Thank you for your valuable comment. The total fabrication cost for the entire process of the 2D-patterned Cu-SHBL-SLP collector has been carefully calculated, detailed information as seen in **Supplementary Table S8**. A single 2D-patterned Cu-SHBL-SLP collector (active area $\approx 0.646 \text{ cm}^2$) costs approximately 3.88 yuan (\$ 0.54) and achieves a water collection rate of $61588 \text{ mg cm}^{-2} \text{ h}^{-1}$ (fog flow at 1500 mL/h). Consequently, this corresponds to a remarkably low water production cost of ≈ 0.006 yuan (\$ 0.0008) per liter per square meter per hour (fog flow at 1500 mL/h), demonstrating excellent cost efficiency and scalability. For large-scale applications, the total fabrication cost of the 2D-patterned Cu-SHBL-SLP is \$ 76.6 per 1 m^2 . A detailed discussion can now be found on **Page 10 Line 9**. To construct stable SLIPS, we employed PFPE lubricant, which is relatively expensive. Developing low-cost stable lubricants represents a promising research direction for practical fabrication and application, contributing to the development of highly efficient bio-inspired fog collectors. It is important to note that our current work presents a proof-of-concept demonstrating the multi-bioinspired fog collection approach; subsequent efforts focused on cost reduction and efficiency improvement are warranted. We would like to appreciate for your valuable suggestions once again.

Supplementary Table S8. Summary of the chemical reagents and materials cost for the fabrication of Cu-SHBL-SLP films.

Materials	Prices	Amount	Cost
Red copper sheet	\$ 12.1/m ²	1 m ²	\$ 12.1
NaOH	\$ 3.4/500g	0.680 kg	\$ 4.6
(NH ₄) ₂ S ₂ O ₈	\$ 4.2/500g	0.204 kg	\$ 1.7
PFTS	\$ 12.1/25g	0.011 kg	\$ 5.3
BaCl ₂	\$ 5.6/500g	0.710 kg	\$ 7.9
K ₂ SO ₄	\$ 4.6/500g	0.520 kg	\$ 4.7
PFPE	\$ 42.0/25g	0.024 kg	\$ 40.3
Total	-	-	\$ 76.6

Reviewer 2:

Comment: In this work the authors present a new method for the fabrication of biphilic SLIP surfaces using top interfacial growing of BaSO₄ nanoparticles to develop hydrophilic tips on Cu(OH)₂ nanorods, followed by a very smart underwater lubricant infusion of the substrate, which allows the preservation of the hydrophilicity of the tips. This is a new part in terms of fabrication. Another smart and new element was the fabrication of 3D channels on such surfaces, which was also presented, followed by the fabrication of a prototype device. The samples were used for the application of fog collection, providing high collection rates, although comparisons with the literature cannot be conclusive as I will explain below. In general, the paper is very interesting with significant novelty and the characterization is detailed and accompanied by simulations, although this is a part that I am not an expert to discuss. I am satisfied with the fabrication and characterization. However, I find that there are some misconceptions and some potentially not fully justified comparisons with the literature, which if not clarified significantly undermine the value of this work in any journal that it may be published. Below you can find comments that can improve the clarity of the manuscript and address these points.

Response: We sincerely appreciate your valuable time and constructive feedback on our manuscript. We have carefully addressed all points raised, especially the calculation of the efficiency factor, as details in the response to **Q3**. The revised manuscript has been submitted, and we believe it now comprehensively incorporates these suggestions, resulting in improved readability.

Q1. The authors mention that the interplay between nucleation and transportation is the key factor for fog collection. However the authors use fog collection. There is a common misconception at the literature of atmospheric water harvesting between the two collection methods, fog and dew collection. During fog collection, a mixture of air and water droplets come in contact with the collection device and the water is being captured due to the adhesion of it with the surface, in that case there is no nucleation or condensation, as there is no phase change of water vapor to liquid vapor (as in the case of dew). To that extent, the authors should revise their manuscript to correct this narrative. Therefore, I would suggest to study more carefully related works such as those below to understand the differences and avoid any confusion or misinterpretation. This is a comment related to the introduction not the experimental work, and thus it does not affect the validity of the results presented, but it should be clearly addressed in the introduction.

Lu, H.; Shi, W.; Guo, Y.; Guan, W.; Lei, C.; Yu, G. Materials Engineering for Atmospheric Water Harvesting: Progress and Perspectives. *Adv. Mater.* 2022, 34 (12), 1–16. <https://doi.org/10.1002/adma.202110079>. That you have already included in your references, as well as Nioras, D.; Ellinas, K.; Gogolides, E. Atmospheric Water Harvesting on Micro-Nanotextured Biphilic Surfaces. *ACS Appl. Nano Mater.* 5 (8), 11334–11341. <https://doi.org/10.1021/acsnm.2c02439>. And Seo, D.; Lee, J.; Lee, C.; Nam, Y. The Effects of Surface Wettability on the Fog and Dew Moisture Harvesting Performance on Tubular Surfaces. *Sci. Rep.* 2016, 6 (March), 1–11. <https://doi.org/10.1038/srep24276>.

Response: Thank you for your thoughtful guidance and valuable suggestions. In this work, our methodology eliminates the need for additional refrigeration equipment to lower the sample surface temperature, meaning that water collection occurs exclusively through fog capture rather than a continuous vapor-to-liquid phase change. In our system, water droplets are selectively captured on the hydrophilic regions due to their strong interaction and adhesion with these sites, which initiates subsequent transportation.

As you rightly pointed out, the term "**initial nucleation**" was inaccurately used in this context and has been corrected to "**initial water capture**" to more accurately describe the process. We have carefully reviewed the references

you provided, including Lu et al. (2022), Nioras et al. (2022), and Seo et al. (2016), to better clarify the distinctions between fog and dew collection. Accordingly, we have made revisions to the introduction as well as the whole manuscript, and cited these references. Please refer to the abstract and introduction section in the revised manuscript for the updated descriptions. We sincerely appreciate your constructive feedback, which has helped us refine the narrative, improve clarity, and enhance the scientific rigor of our work. Thank you again for your support.

Q2. Fog collection results at figure 2a) are the most important to explain fog collection mechanism on the newly introduced surfaces compared to uniform wettability ones, thus they should be analyzed in detail in text as well, explaining the reason of the differences between the surfaces.

Response: Thank you for your valuable suggestion. Following your recommendation, we have provided a more detailed analysis of the fog collection results shown in Figure 2a in the revised manuscript.

As depicted in Figure 2a, we systematically analyzed the fog collection performance of various surfaces with distinct wettability characteristics. The pristine Cu film, with its intrinsic hydrophobicity, exhibited a water collection rate (WCR) of $896.8 \text{ mg cm}^{-2} \text{ h}^{-1}$ and a water collection efficiency (η) of 8.45%; The Cu-SHL film demonstrated the lowest performance (WCR: $844.0 \text{ mg cm}^{-2} \text{ h}^{-1}$, η : 7.95%), which can be attributed to its superhydrophilic surface. This extreme wettability leads to the formation of a dense water film, significantly impeding fog capture and droplet removal. The Cu-SHB film, with superhydrophobic modification, exhibited a higher performance (WCR: $1187.9 \text{ mg cm}^{-2} \text{ h}^{-1}$, η : 11.20%) due to its water-repellent properties and improved droplet transport dynamics. The integration of hydrophilic components in the hybrid Cu-SHBL film further improved the WCR to $1317.8 \text{ mg cm}^{-2} \text{ h}^{-1}$ and η to 12.40%, as the hydrophilic regions enhanced fog capture capability. The Cu-SHBL-SLP film demonstrated the best performance (WCR: $1663.7 \text{ mg cm}^{-2} \text{ h}^{-1}$, η : 15.68%). This significant enhancement is attributed to the presence of exposed hydrophilic bumps combined with the ultra-low sliding angle and minimal frictional resistance of the bioinspired slippery liquid-infused porous surface (SLIPS), which synergistically optimize both fog capture and droplet removal.

This detailed discussion has been incorporated into the revised manuscript (see **Page 6 Lines 23**). We sincerely appreciate your insightful feedback, which has allowed us to refine our explanation and provide a more comprehensive analysis of the results. Thank you again for your suggestion.

Q3. In Figure 4 e) and in supplementary data the authors make an attempt to compare the results of the literature with this work on WCR, based on the fog flow in each experiment. Although this graph is a good indication for the general state of the fog collection, and the rather good results of this work, there are many more factors that contribute to the WCR, such as the distance from the fog source, the relative sample size to the fog flow projection e.t.c. Therefore it may be that comparisons are not always easy or "fair". The following work has introduced an efficiency factor in order to allow fair comparison of literature results in different setups. Perhaps, using an efficiency factor such comparison may enhance even more or reduce the performance of the surfaces introduced in this paper.

Response: Thank you for your professional and insightful suggestion. Following your recommendation, to provide a more objective assessment of the water collection performance and ensure fair comparisons with literature results, we have introduced the water collection efficiency (η) as a key evaluation metric¹. This efficiency is defined as the ratio of the experimental water collection rate (W_{exp}) to the theoretical water collection rate (W_{the}), expressed as follows:

$$\eta = \frac{W_{exp}}{W_{the}} \times 100\%$$

where W_{exp} and W_{the} are the measured experimental water collection rate and calculated theoretical water collection rate of the samples. The effect of experimental and setup conditions could be eliminated. The calculation of maximum available fog for collection follows the turbulent jet theory presented by Cushman-Roisin². In this work, fog flow with 300 mL/h was outputted by a nozzle with the diameter of 2.2 cm, and the Reynolds number was calculated as 15181, which was termed as turbulent flow and suitable for applying the general theory. The theoretical water collection rate (W_{the}) could be calculated as follows:

$$W_{the} = u_{max} \times c_{max}$$

where u_{max} and c_{max} were the fog velocity and fog concentration at the sample surface. As the fog jet propagates downstream from the nozzle, its velocity and absolute humidity exhibit progressive decay, as follows:

$$u(x, r) = \int u_{max}(x) e^{-50r^2/x^2} dr$$

$$c(x, r) = \int c_{max}(x) e^{-50r^2/x^2} dr$$

Velocity and humidity concentration exhibit non-uniform distributions across the sample surface, with both parameters following Gaussian-like profiles that peak centrally, as illustrated below:

$$u_{max}(x) = u_0 \times \frac{5d}{x}$$

$$c_{max}(x) = c_0 \times \frac{5d}{x}$$

where u_0 and c_0 are the velocity and concentration of the nozzle jet, and the variable x quantifies the distance between sample surface and the jet cone apex. In this work, x value of the low-powered ultrasonic humidifier (YADU) with the fog flow at 300 mL/h was around 30 cm, and the high-powered ultrasonic humidifier (AUX) with the fog flow of 420-1500 mL/h was around 20 cm. Meanwhile, the air was assuming saturated with moisture at 20 °C, and the c_0 value was 17 g/m³. The velocity could be calculated as follows:

$$u_0 = \frac{Q}{S}$$

where S was the nozzle area. In this work, the nozzle diameter $d=2.2$ cm, and the S was calculated as 3.8 cm². When the low-powered ultrasonic humidifier was applied, the carrier gas flow rate (Q) could be calculated as 294.2 L/min, and the u_0 could be calculated as 12.90 m/s. Therefore, the W_{the} could be calculated as 10610.3 mg cm⁻² h⁻¹, and all the water collect efficiency (η) could be calculated according to the measured experimental water collection rate (W_{exp}). For example, the 2D-patterned Cu-SHBL-SLP film exhibited the highest performance, achieving a water collection rate (WCR) of 5081.25 mg cm⁻² h⁻¹ and a water collection efficiency (η) of 47.89%. This significant improvement is primarily attributed to the unique water transport mechanism on the surface, driven by Laplace pressure, which enhances both fog capture and droplet removal. The detailed calculations and results for water collection efficiency (η) across all tested samples are now included in the revised manuscript (**Page14 Line9**, and **Supplementary Table S1, S3, S4, S5, S7**). We sincerely appreciate your suggestion, which has allowed us to adopt a more rigorous approach to evaluating and comparing performances. Thank you for helping us enhance the scientific rigor and objectivity of our work.

Supplementary Table S1. Summary of water collection performance of different samples. Theoretical water collection rate ($\text{mg cm}^{-2} \text{h}^{-1}$), experimental water collection rate ($\text{mg cm}^{-2} \text{h}^{-1}$) and water collection efficiency (%) of original Cu, Cu-SHL, Cu-SHB, Cu-SHBL and Cu-SHBL-SLP film.

Sample	Theoretical water collection rate ($\text{mg cm}^{-2} \text{h}^{-1}$)	Experimental water collection rate ($\text{mg cm}^{-2} \text{h}^{-1}$)	Water collection efficiency (%)
Cu	10610	897	8.45
Cu-SHL	10610	844	7.95
Cu-SHB	10610	1188	11.20
Cu-SHBL	10610	1318	12.42
Cu-SHBL-SLP	10610	1664	15.68

Supplementary Table S3. Summary of water collection performance of different samples. Theoretical water collection rate ($\text{mg cm}^{-2} \text{h}^{-1}$), experimental water collection rate ($\text{mg cm}^{-2} \text{h}^{-1}$) and water collection efficiency (%) of Cu-SHB-SLP, Cu-SHBL-0.27%-SLP, Cu-SHBL-0.36%-SLP, Cu-SHBL-0.53%-SLP, Cu-SHBL-0.87%-SLP and Cu-SHBL-1.54%-SLP film.

Sample	Theoretical water collection rate ($\text{mg cm}^{-2} \text{h}^{-1}$)	Experimental water collection rate ($\text{mg cm}^{-2} \text{h}^{-1}$)	Water collection efficiency (%)
Cu-SHB-SLP	10610	1250	11.78
Cu-SHBL-0.27%-SLP	10610	1306	12.31
Cu-SHBL-0.36%-SLP	10610	1510	14.23
Cu-SHBL-0.53%-SLP	10610	1664	15.68
Cu-SHBL-0.87%-SLP	10610	1508	14.21
Cu-SHBL-1.54%-SLP	10610	1382	13.02

Supplementary Table S4. Summary of water collection performance of different samples. Theoretical water collection rate ($\text{mg cm}^{-2} \text{h}^{-1}$), experimental water collection rate ($\text{mg cm}^{-2} \text{h}^{-1}$) and water collection efficiency (%) of 2D patterned Cu-SHBL-SLP film with different L values.

Sample	Theoretical water collection rate ($\text{mg cm}^{-2} \text{h}^{-1}$)	Experimental water collection rate ($\text{mg cm}^{-2} \text{h}^{-1}$)	Water collection efficiency (%)
Cu-SHBL-SLP (L=0.5 cm, $\theta=60^\circ$)	10610	4546	42.85
Cu-SHBL-SLP (L=1.0 cm, $\theta=60^\circ$)	10610	5081	47.89
Cu-SHBL-SLP (L=1.5 cm, $\theta=60^\circ$)	10610	4561	42.89
Cu-SHBL-SLP (L=2.0 cm, $\theta=60^\circ$)	10610	4146	39.07
Cu-SHBL-SLP (L=2.5 cm, $\theta=60^\circ$)	10610	3696	34.83

Supplementary Table S5. Summary of water collection performance of different samples. Theoretical water collection rate ($\text{mg cm}^{-2} \text{h}^{-1}$), experimental water collection rate ($\text{mg cm}^{-2} \text{h}^{-1}$) and water collection efficiency (%) of 2D patterned Cu-SHBL-SLP film with different θ values.

Sample	Theoretical water collection rate ($\text{mg cm}^{-2} \text{h}^{-1}$)	Experimental water collection rate ($\text{mg cm}^{-2} \text{h}^{-1}$)	Water collection efficiency (%)
Cu-SHBL-SLP (L=1.0 cm, $\theta=30^\circ$)	10610	4941	46.57
Cu-SHBL-SLP (L=1.0 cm, $\theta=60^\circ$)	10610	5081	47.89
Cu-SHBL-SLP (L=1.0 cm, $\theta=90^\circ$)	10610	4461	42.04
Cu-SHBL-SLP (L=1.0 cm, $\theta=120^\circ$)	10610	4129	38.92
Cu-SHBL-SLP (L=1.0 cm, $\theta=150^\circ$)	10610	3355	31.62

Supplementary Table S7. Summary of the reported water collection performance. Fog flux (mL/h), theoretical water collection rate ($\text{mg cm}^{-2} \text{h}^{-1}$), experimental water collection rate ($\text{mg cm}^{-2} \text{h}^{-1}$) and water collection efficiency (%) of reported works.

Sample	Fog flow (mL/h)	Theoretical water collection rate ($\text{mg cm}^{-2} \text{h}^{-1}$)	Experimental water collection rate ($\text{mg cm}^{-2} \text{h}^{-1}$)	Water collection efficiency (%)	Refs
Long plasma-textured PMMA	25.2	4700	1171	25	1
PFC	1080	11800	4210	35.68	3
Cu-SHBL-SLP	300	10610	5081	47.89	This work
Cu-SHBL-SLP	420	33423	11238	33.63	This work
Cu-SHBL-SLP	900	71620	25207	35.20	This work
Cu-SHBL-SLP	1500	119366	61588	51.60	This work

References

- 1 Nioras, D., Ellinas, K., Constantoudis, V. & Gogolides, E. How Different Are Fog Collection and Dew Water Harvesting on Surfaces with Different Wetting Behaviors? *ACS Appl Mater Interfaces* **13**, 48322-48332, doi:10.1021/acsami.1c16609 (2021).
- 2 Cushman-Roisin, B. *Environmental Fluid Mechanics*; John Wiley & Sons, Inc, 2013; pp 141–147.
- 3 Yan, D., Chen, Y., Liu, J. & Song, J. Super - Fast Fog Collector Based on Self - Driven Jet of Mini Fog Droplets. *Small* **19**, doi:10.1002/smll.202301745 (2023).

Reviewer 3

In their manuscript "Preserving exposed hydrophilic bumps on multi-bioinspired slippery surface arrays unlocks high-efficiency fog collection and cleaning", the authors describe the design, manufacturing and testing of a fog collection device. The function of the device combines a hydrophilic tip with a slippery hydrophobic surface in order to allow faster drop collection by focusing drop growth to small hydrophilic zones. I was asked to comment only on the MD simulations. The actual device describes superhydrophobic nanorods infused in an oil film. The key facts amenable to MD simulation are droplet nucleation and growth. One might also try to simulate slippage or the manufacturing process. The MD results show a system made of PFPE and a BaSO₄ substrate. It should be noted that the actual nanorods are made of (functionalized) Cu(OH)₂ and are not modeled, however the main problem is clearly the PFPE lubricating oil. Because the MD simulation is very short (0.5 ns), and the water concentration is very high, the PFPE does not have the time to equilibrate into a proper hydrophobic film. Instead, we observe droplet formation inside the PFPE (3f), which should simply not happen. Obviously, the system must be equilibrated before adding the water film. It might even be that the force field parameters predict hydrophilic PFPE, which would mean they are not adequate. The ultra-short timescale at high water content and low temperature just shows that the instantaneously formed droplet attaches itself to the BaSO₄, in other words, that the BaSO₄ surface is indeed hydrophilic. There has been no serious attempt to study of nucleation rate of droplets and wetting behavior at appropriate conditions. If this was done, the simulation might be more convincing. However, the core of the paper is not the discovery that BaSO₄ crystals will bind to water droplets or that PFPE is hydrophobic. If anything it is about the "trick" to keep them exposed by performing the manufacturing process in water, i.e. keeping the BaSO₄ tip wet while oiling the nanorods. I think this process would be a more interesting thing to study. Alternatively, continuum modelling could be performed, if surface tensions were known, of the droplet binding and slippage behavior. Furthermore, the computational details contain some trivial "equations" for LJ and Coulomb potentials, a reference to LAMMPS and mentioning materials studio. Instead of the equations, details on the potentials used should be given. To make the simulation reproducible, the precise parameters used for water, PTFE and BaSO₄ should be stated (in the SI) or referenced. If new parameters were used, a validation procedure will have to be reported as well. The MD details also miss references to treatment of boundary conditions (periodic electrostatics? Barrier in z?) and the correct references of the methods used. Referencing classical MD is not necessary and the reference is not to the first use of the method. These are beginner's mistakes. Despite the pretty images, I do not think these simulations add any scientific value at present and should be removed or completely reworked. The recommendation "do not publish" pertains only to the MD data.

Response: We sincerely thank the reviewer for their insightful and detailed feedback on the molecular dynamics (MD) simulations. We fully acknowledge the limitations of our current MD results, which are unable to adequately address the scientific questions posed in this study.

We greatly appreciate the reviewer's expertise in identifying several critical issues, including the insufficient equilibration time, the inappropriate simulation conditions, and the lack of proper force field validation. We also acknowledge the need for more detailed computational parameters, improved methodology, and a rigorous analysis of droplet nucleation and wetting behavior under relevant conditions. Furthermore, we deeply regret the inclusion of incomplete computational details and the overly simplistic equations, which are indeed beginner-level errors.

After careful consideration, and in light of the reviewer's constructive critique, we have decided to **remove the MD simulations entirely** from the manuscript. Given that the primary focus of this work lies in the design, fabrication, and experimental demonstration of the fog collection device, we believe that removing the simulations will allow the

manuscript to maintain its scientific rigor and focus on the robust experimental findings. We are immensely grateful for the reviewer's guidance, which has helped us to critically evaluate and refine our work. Thank you again for your time and expertise.

Response to Reviewers

Reviewer 1:

Comment: This study was conducted with many experiments and detailed analyses, and through meticulous commentary and supplementation, I agree to include it in this journal.

Response: Thank you so much for your positive feedback and for recognizing the effort put into our study. We are truly grateful for your meticulous commentary and valuable suggestions, which have greatly contributed to enhancing the quality of our manuscript. We sincerely appreciate your time and support in agreeing to include our work in the journal.

Reviewer 2:

Comment: The authors have adequately addressed the comments and questions raised in the first review, and therefore I recommend the publication of the article. One suggestion is that the water collection efficiency calculations could be moved to the Supplementary Information to make the main text more concise and easier to read.

Response: We are truly grateful for the reviewer's supportive feedback and valuable recommendation for publication. We especially appreciate thoughtful suggestion regarding the placement of the water collection efficiency calculations, as we agree that maintaining conciseness in the main text is important. After careful consideration and following discussions with the editorial team, we have opted to retain this section in the main manuscript to ensure immediate accessibility of these fundamental results for readers. Nevertheless, we have significantly refined and condensed the relevant subsection to improve readability and focus, which was an excellent outcome inspired by the reviewer's comment. We thank the reviewer again for raising this point, as it helped us improve the overall balance and clarity of the manuscript.